# Transient Global Amnesia (TGA): Sex-Specific Differences in Blood Pressure and Cerebral Microangiopathy in Patients with TGA

**DOI:** 10.3390/jcm11195803

**Published:** 2022-09-30

**Authors:** Andreas Rogalewski, Anne Beyer, Anja Friedrich, Frédéric Zuhorn, Randolf Klingebiel, Friedrich G. Woermann, Sabine Oertelt-Prigione, Wolf-Rüdiger Schäbitz

**Affiliations:** 1Department of Neurology, Evangelisches Klinikum Bethel, University Hospital OWL, University Bielefeld, 33611 Bielefeld, Germany; 2Department of Psychology, Bielefeld University, 33615 Bielefeld, Germany; 3Department of Neuroradiology, Evangelisches Klinikum Bethel EvKB, University Hospital OWL, University Bielefeld, 33617 Bielefeld, Germany; 4Department of Epileptology (Krankenhaus Mara), University Hospital OWL, University Bielefeld, 33617 Bielefeld, Germany; 5Medical Faculty OWL, Bielefeld University, 33617 Bielefeld, Germany

**Keywords:** transient global amnesia, gender, sex, cerebral microangiopathy, hypertension, blood pressure, risk factor

## Abstract

Transient global amnesia (TGA) is defined by an acute memory disturbance of unclear aetiology for a period of less than 24 h. Observed psychological, neuroanatomical and hormonal differences between the sexes in episodic memory suggest sex-specific differences in memory disorders such as TGA. The aim of this study was to determine sex-specific differences in cardiovascular risk profiles, recurrences and magnetic resonance imaging (MRI). In total, 372 hospitalised TGA patients between 01/2011 and 10/2021 were retrospectively analysed. Comparisons were made between female and male TGA patients and compared to 216 patients with acute stroke. In our sample, women were overrepresented (61.8%), especially compared to the general population in the 65–74 age category (χ^2^ = 10.6, *p* < 0.02). On admission, female TGA patients had significantly higher systolic blood pressure values and a higher degree of cerebral microangiopathy compared to male TGA patients, whereas acute stroke patients did not. No sex-specific differences were observed with respect to recurrences or hippocampal DWI lesions. Our data demonstrate sex-specific differences in TGA. The higher blood pressure on admission and different degree of cerebral microangiopathy in female TGA patients supports the theory of blood pressure dysregulation as a disease trigger. Distinct precipitating events in female and male patients could lead to differences in the severity and duration of blood pressure abnormalities, possibly explaining the higher incidence in female patients.

## 1. Introduction

Transient global amnesia (TGA) is an acute disturbance of memory function that usually occurs in middle-aged and elderly individuals. It was first described in 1956 [1]. Foreground anterograde and retrograde amnesia without focal neurological deficits are disease hallmarks, lasting for a period of less than 24 hours. This neuropsychological syndrome is thought to be caused by a transient disturbance in hippocampal function [2,3]. In addition, it reflects a reversible disruption of the functional connectivity of the broader episodic memory network, including the medial temporal sub-network, as well as the orbitofrontal-cingulate, inferior temporal, medial occipital, and deep-structure sub-networks [4,5,6].

In general, there is evidence that biological sex plays a crucial role in memory function [7]. In non-demented populations and in both cross-sectional and longitudinal studies, women tend to perform better than men in episodic memory function [7,8]. Both psychological and biological aspects are discussed as possible explanations for the observed sex differences. The affect intensity hypothesis [9], for example, states that the intensity of female reactions to emotional experiences exceeds the average male reactions, thus facilitating the encoding of the memory trace. The cognitive style hypothesis suggests a sex-specific difference in the detail of the event encoding, with a different degree of lateralisation as a possible cause [10]. Furthermore, hormonal differences, in particular a correlation of higher estrogen levels with greater episodic memory performance [11], differential activation of parahippocampal regions [12], and observed neuroanatomical sex differences with greater volumes of hippocampal tissue (relative to brain size) [13,14] might be possible explanations.

The observed psychological, neuroanatomical and hormonal sex differences in episodic memory suggest a role in sex differences with regard to dysfunctional performance, for example, in TGA. A recently published retrospective observational study showed a predominance of females regarding the incidence of TGA and a sex-specific difference in various precipitating events in TGA patients, with more emotional triggers in females and more physical triggers in males [15].

Although other studies also reported a predominance of females in TGA occurrence [16,17,18,19], a corresponding meta-analysis did not identify significant sex differences between the 1333 patients enrolled in 52 published case series (*n* = 91) and 34 published group studies (*n* = 1171) [19]. Yet, differences in triggering events were also reported in a recent review article [20]. Male patients more often reported physical precipitating events and chronic hypertension, while female TGA patients more frequently experienced emotional precipitating events, emotional and personality disorders, a positive history of migraine and a younger age of onset [20].

Overall, sex differences played only a minor role in the majority of TGA studies.

The aim of the present study was to analyse sex-specific differences of TGA patients in a large retrospective cohort of a tertiary referral centre. In previous studies, we were able to show differences between TGA patients and acute stroke patients as well as differences between TGA patients with and without subsequent recurrence [6,21]. It was shown that TGA patients, regardless of sex, had significantly higher blood pressure values on admission compared to acute stroke patients, but were less likely to present with signs of chronic hypertension [21]. Furthermore, the risk of TGA recurrence was lower in younger patients and in cases with an absence of cerebral microangiopathy [6], which is a common cerebral manifestation of chronic hypertension.

The present investigation represents an extension of the cohort from these previously reported studies focussing on sex-specific differences.

## 2. Materials and Methods

### 2.1. Patients

We conducted a retrospective analysis of patient records from a major German hospital (tertiary referral centre). A total of 372 patients with TGA diagnosed by a neurologist according to the criteria defined by Hodges and Warlow (1990) [22] were identified between 1 January 2011 and 30 October 2021. Patients who did not meet the diagnostic criteria were not included in the study. This investigation represents an extension of the cohort from previously reported studies [6,21]. All patients having a discharge diagnosis of TGA during the relevant period were included.

### 2.2. Control Group

In total, 216 consecutive stroke patients served as a reference condition. Patients were included from our Stroke Unit diagnosed with acute ischemic stroke or transient ischemic attack. These data were collected between 24 November 2018, and 3 June 2019. The sample and the data collected originated from a previous study and were not extended by additional cases [21].

### 2.3. Procedure

Patient characteristics were comprehensively assessed as reported previously [21], including demographics, cardiovascular risk factors and neuroradiologic imaging findings (MR or computed tomography [CT] scans, ultrasound, and echocardiography).

The patients’ first blood pressure figures were recorded upon admission. Chronic hypertension was considered as established if already indicated by the patient’s history, or if antihypertensive medication was required because of elevated blood pressure levels during hospital stay and at discharge. Cases normalised initial hypertensive blood pressure peaks during the course of treatment were not considered to be chronic hypertension provided that no permanent antihypertensive medication was required.

MR scans were analysed for signs of stroke and hippocampal diffusion-weighted imaging (DWI) lesions. The extent of cerebral microangiopathy was evaluated by a neuroradiologist using Fazekas’ score (0–3) [23]. The presence of cerebrovascular stenosis was evaluated using neurosonology. Laboratory parameters, including glucose level, HbA1c, cholesterol level, and CRP, were taken from emergency room records. Echocardiography was assessed for septal hypertrophy and left ventricular ejection fraction and septal hypertrophy. Septal hypertrophy was considered a possible indicator of chronic hypertension. The presence of increased septal thickness was defined with the following values: women > 9 mm, men > 10 mm [24]. CHA2DS2-VASc scores were additionally obtained for all patients at the time of discharge. Antihypertensive medication at discharge was considered as hypertension in the calculation of the CHA2DS2-VASc score. Female sex was not considered in the CHA2DS2-VASc scores, as this would have led to a false positive difference in the sex-dependent analysis.

### 2.4. Data Analysis

Data analysis was carried out using Statistical Package for the Social Sciences (SPSS) version 25 2018 (IBM, Armonk, NY, USA). Descriptive statistics were displayed as mean ± standard deviation for continuous data and frequencies with percentages for categorical variables. Normal distribution of residuals was assessed via Shapiro–Wilk test with *p* < 0.05 indicating non-normal distribution. Homoscedasticity was assessed visually via q-q-plots.

Demographic characteristics (age, sex) were compared by using parametric *t*-tests or non-parametric Mann–Whitney U tests, depending on normal distribution. In order to contrast our sample characteristics to the demographic characteristics of the population in our catchment area, publicly available population statistics of age and sex distributions [25] were obtained and compared to our sample via chi-square tests.

In line with the analysis of demographic characteristics, sex differences in the profile of cardiovascular risk factors in TGA patients were assessed by using parametric *t*-tests or non-parametric Mann–Whitney U tests for ordinal and interval dependent variables (e.g., blood pressure, serum glucose levels, HbA1c). Sex differences in categorical variables (e.g., hypertension, diabetes mellitus, and hypercholesterolemia) were assessed with chi-square tests. A significance level of less than 0.05 in the two-sided test was assumed to be significant.

Overall, more than 20 separate statistical tests were computed, as more sophisticated statistical methods (e.g., MANOVA, structural equation modelling) were infeasible due to mostly categorical outcome variables and requirement violations. In order to correct for alpha error accumulation in multiple testing, p-values were adjusted using the Bonferroni method (*p*_adj_ = *p*_obs_ × k; where *p*_adj_: adjusted *p*-value; *p*_obs_: observed *p*-value; k = number of comparisons) [26]. To prevent overcorrection and to preserve statistical power, the number of comparisons (k) was determined for each set of outcome variables separately: blood pressure (k = 2: systolic and diastolic blood pressure), blood glucose (k = 2: HbA1c, serum glucose level), and MR imaging abnormalities (k = 2: presence of DWI lesion, unilateral versus bilateral lesion in case of presence of DWI lesion).

## 3. Results

### 3.1. Demographic Characteristics

In total, 372 TGA patients were included in this study. Female patients dominated the study cohort (61.8%), but without any statistically significant age difference between the two groups (women: 68.1 ± 9.6 years versus men: 65.9 ± 11.4; *p* = 0.167). Further characteristics are presented in Table 1.

A comparison of observed and expected frequencies in the different age groups revealed significant differences in terms of the sex-specific distribution in the various age categories. Specifically, women aged 65–74 years were relatively overrepresented in our study population (χ^2^ = 10.6, *p* < 0.02). Due to small case numbers (*n* = 6), patients younger than 45 years were not included in this calculation. The comparison of the age distribution within the various age categories is shown in Figure 1.

### 3.2. Vascular Risk Factors

Analysis of vascular risk factors in TGA patients on admission, in regard to sex-specific differences, showed significant higher systolic blood pressure (SBP) figures in female patients (173.2 ± 23.4 mm Hg versus 165.8 ± 22.0 mm Hg; *p* = 0.007), increased serum cholesterol levels (221.6 ± 40.7 mg/dL versus 207.6 ± 45.5 mg/dL; *p* = 0.005) as well as augmented C-reactive protein (CRP) levels (2.8 ± 6.4 mg/L versus 1.7 ± 1.8 mg/L; *p* = 0.011).

There were no differences in other vascular risk factors on admission (diastolic blood pressure, HbA1c, CHA_2_DS_2_-VASc score) between female and male TGA patients, as assessed by parametric *t*-test (in case of normal distribution: diastolic blood pressure) or Mann–Whitney U tests (in case of non-normal distribution: other parameters) (see Figure 2 and Table 1). Apart from this, no differences in cardiovascular comorbidity were detected in regard to chronic hypertension, intake of antihypertensive drugs at discharge, diabetes mellitus, septal hypertrophy in transthoracic echocardiography, presence of cerebrovascular stenosis, atrial fibrillation, previous stroke, and left ventricular ejection fraction (LVEF) in transthoracic echocardiography (see Table 1).

### 3.3. Comparison with Acute Stroke Patients

As a control group, the significant different results (systolic blood pressure values, cholesterol levels, and CRP levels) were analysed in a cohort from an earlier published study consisting of 216 consecutive patients of our Stroke Unit with acute ischemic stroke or transient ischemic attack collected between 24 November 2018, and 3 June 2019 [21]. The inclusion criteria and demographic characteristics of the collective can be found in the cited publication. After statistical correction for age, the mentioned study showed that TGA patients had higher systolic blood pressure, higher cholesterol levels, and lower CRP levels than acute stroke patients. Supplementary analyses of these acute stroke patients were now carried out under the aspect of sex-specific differences of these parameters as a control condition of our significant parameters. The values for systolic blood pressure and cholesterol levels were not normally distributed, so the comparison was carried out using the Mann–Whitney U test. The CRP values were normally distributed, and therefore the comparison was made using a parametric *t*-test.

The analysis of acute stroke patients on admission in regard to sex-specific differences showed no differences of systolic blood pressure figures between the sexes (female stroke patients 162.6 ± 24.5 mm Hg, male stroke patients 161.1 ± 28.6 mm Hg, Mann–Whitney U 4565.000, Z = 0.473, *p* = 0.636). Serum cholesterol levels were higher in female stroke patients compared to male stroke patients (205.9 ± 52.0 mg/dL versus 173.7 ± 42.0 mg/dL; Mann–Whitney U 7162.000, Z = 3.420, *p* = 0.001). CRP levels showed a not significant tendency to be higher in female stroke patients compared to male stroke patients (14.8 ± 30.1 mg/L versus 8.2 ± 19.1 mg/L; T = 1.857, *p* = 0.065).

While TGA patients presented sex-specific differences in systolic blood pressure values on admission, this was not the case in acute stroke patients. The sex-specific difference in cholesterol levels in TGA patients was also observed in acute stroke patients, but only trended in CRP levels.

### 3.4. TGA Recurrences

In total, 29 patients suffered a TGA relapse with subsequent treatment in our hospital. All subsequent inpatient and outpatient contacts were analysed for these patients to identify possible recurrences, even if the patients were taken to the emergency department or outpatient department for other complaints. Recurrence assessment by a neurologist was likewise performed, using the diagnostic TGA criteria. Our hospital operates the only department of neurology in a catchment area of about 320,000 people. We therefore assumed that patients with TGA symptoms within this area most probably presented to the emergency room of our institution [6]. Regarding the frequency of recurrences, no sex-specific differences were detected in our study (8.3% in female TGA patients 8.3% versus 7.0% in male TGA patients; χ^2^ = 0.181, *p* = 0.670) (see Table 1).

### 3.5. MR Imaging Abnormalities

Imaging findings in 205 (of 230) female TGA patients and 122 (of 142) male TGA patients, who underwent a high-field MRI (1.5 and/or 3.0 Tesla), were evaluated. There were no significant differences between both groups in the presence and laterality (unilateral versus bilateral) of hippocampal DWI lesions (see Table 1).

The extent of cerebral microangiopathy was assessed using Fazekas’ score. Fazekas’ scoring excluded cerebral microangiopathy (Fazekas’ score 0) in 30.7% of female TGA patients (42.1% of male TGA patients), and disclosed microvascular findings as follows: mild microangiopathy (score 1) in 52.7% (47.9%), moderate microangiopathy (score 2) in 13.2% (7.4%), and severe microangiopathy (score 3) in 3.4% (2.5%). The distribution of microangiopathic lesions using Fazekas’ score in both groups (female versus male TGA patients) is displayed in Figure 3. Using the Mann–Whitney U test, the degree of microangiopathy (ranking from 0 = none to 3 = severe) was significantly lower in male TGA patients compared to female TGA patients (U = 14,147.500, z = 2.338, *p* = 0.019). The effect size according to Cohen was r = 0.22, corresponding to a small effect.

### 3.6. Comparison with Acute Stroke Patients

As a control group, the distribution of the degree of cerebral microangiopathy was also analysed with regard to sex-specific differences in the cohort of acute stroke patients from our earlier published study [21]. Fazekas’ scoring excluded cerebral microangiopathy (Fazekas’ score 0) in 30.2% of female TGA patients (21.5% of male TGA patients), and disclosed microvascular findings as follows: mild microangiopathy (score 1) in 20.8% (35.4%), moderate microangiopathy (score 2) in 26.4% (20.0%), and severe microangiopathy (score 3) in 22.6% (23.1%). Using Mann–Whitney U test, the degree of microangiopathy (ranking from 0 = none to 3 = severe) was different between the sexes in acute stroke patients (U = 1691.500, z = 0.173, *p* = 0.862).

While TGA patients showed sex-specific differences in the distribution of the degree of cerebral microangiopathy, this was not the case in acute stroke patients.

## 4. Discussion

The present study aimed at identifying sex-specific differences of TGA patients in cardiovascular risk profiles, recurrences, and MRI. In total, 372 hospitalised TGA patients between 01/2011 and 10/2021 were retrospectively evaluated, comparing female and male TGA patients.

Females were overrepresented in our sample (61.8%), but the age-adjusted prevalence rates did not differ between the sexes. The sample displayed a slight overrepresentation of women compared to the general population in the 65–74 age category.

Female TGA patients displayed significant higher systolic blood pressure values, augmented serum cholesterol levels and increased C-reactive protein levels on admission compared to male TGA patients, even though no age difference between the groups existed. Furthermore, female TGA patients displayed a greater extent of cerebral microangiopathy than male TGA patients.

The most interesting finding of our study is the significantly higher blood pressure of female TGA patients compared to males upon admission to hospital. Previous work by our group suggested that that TGA patients, regardless of sex, had significantly higher blood pressure values on admission compared to acute stroke patients, but were less likely to present with signs of chronic hypertension [21]. Furthermore, the risk of TGA recurrence was lower in younger patients when cerebral microangiopathy was absent [6].

The significant different blood pressure levels can be discussed in the context of differing precipitating events prior to TGA in the different sexes. Male patients show a more frequent association between TGA and physical events [15,20]. Females, on the other hand, show a stronger relationship with emotional events, emotional and personality disorders, positive history of migraine and younger age of onset [15,20]. This association has been repeatedly described in various studies and was also discussed in a recent review [20]. A similar pattern has also been attributed to precipitating events in myocardial infarction [27,28,29]. This phenomenon, commonly referred to as “mental stress-induced myocardial ischemia” (MSIMI) has a prognostic value similar to that of exercise or pharmacologically induced ischemia, but it occurs at lower levels of oxygen demand than exercise-induced ischemia, and is not related to the severity of coronary artery disease or previous revascularisation [28]. However, vessel differences, such as cerebrovascular autoregulation, precludes the simple transfer of coronary to cerebral pathophysiologic principles.

The pathophysiology of blood pressure crises due to acute mental stress is quite different compared to blood pressure crises due to physical activity. Acute mental stress causes the activation of the sympathetic nervous system as well as the limbic system with the subsequent activation of the sympathetic nervous system and the sympathetic adrenal medullary system via the release of catecholamines and cortisol [30]. Normalisation occurs slowly via a negative feedback loop of rising catecholamine and cortisol levels via the inhibition of corticotropin releasing factor and adrenocorticotropic hormone production, causing blood pressure to return to baseline [31]. It is known that it takes hours for cortisol levels to return to baseline levels despite the previous normalisation of behaviour [32]. Sex differences have been established in stress-related hormonal secretion, with females having increased hormonal secretion in comparison to males regardless of age [33]. Interestingly, psychological stress can additionally influence the tightness of the blood–brain barrier [34] and thus may affect the functionality of vulnerable brain structures such as the hippocampal region in TGA patients. In contrast to mental stress, physical activity leads to a short-term increase in blood pressure and normalisation within minutes after the end of the activity [35]. Depending on the physical training status and pre-existing chronic hypertension, physical activity partially leads to lower blood pressure values post-exercise compared to immediately pre-exercise measured blood pressure. This pathophysiological understanding and the previously reported sex-specific differences in precipitating events before TGA may explain why females had elevated blood pressure levels on admission compared to males.

At this point, it may be postulated that different mechanisms between females and males trigger TGA. Females show more frequent association with emotional events, emotional and personality disorders [15,20]. In this context, acute mental stress reactions trigger hormonally induced, long-lasting hypertensive episodes over hours [30,31,32], affecting the blood–brain barrier [34] and affecting the LTP of the hippocampal CA-1 region [36]. These prolonged blood pressure episodes may further explain the observed higher extent of cerebral microangiopathy in females. 

In contrast, males are more likely to have exertion-associated blood pressure abnormalities with functional interactions of angiotensin II type 1 and N-methyl-D-aspartate (NMDAR) receptors [20], so that even shorter-lasting blood pressure elevations could trigger TGA.

This theory is well in line with our previous studies. We were able to show an association between acute hypertension and TGA occurrence [21]. TGA patients suffered less likely from chronic hypertension compared to acute stroke patients. This correlated to less severe cerebral microangiopathy and less septal hypertrophy in transthoracic echocardiography. These observations are supported by evidence of a lower vascular mortality risk in TGA patients compared to TIA patients [37]. We hypothesised that acute hypertensive peaks may trigger TGA episodes, especially in patients not adapted to chronic hypertension [21]. In another study, focusing on risk factors for TGA recurrence, we noticed a higher recurrence risk at younger age, while signs of cerebral microangiopathy were absent [6]. Based on these findings, we concluded that failure of cerebrovascular autoregulation (autoregulatory breakdown) and subsequent hypertensive encephalopathy may play a role in the development of TGA, which occurs at lower blood pressure levels in patients not accustomed to chronic hypertension [21]. The current findings of higher TGA incidence as well as concomitant higher degree of cerebral microangiopathy in females must be interpreted in the light of different precipitating events, as discussed above. Therefore, it is absolutely relevant to screen for precipitating events and for evidence of acute and chronic hypertension in TGA patients [38].

### 4.1. Sex Distribution

Female patients were overrepresented in our study population (61.8%), and were relatively overrepresented when the female to male distribution in the general population of 65–74 year olds in our catchment area is used as a reference. Given that our hospital is the only tertiary referral centre in the area, a relatively higher incidence in female compared to male patients could be postulated. Although other TGA studies also reported a female predominance [16,17,18,19], a corresponding meta-analysis revealed no statistically significant sex difference (46.4% men and 53.6% women; χ^2^ = 0.48, *p* = 0.49) between the 1333 patients recruited from 52 published case series (*n* = 91) and 34 published group studies (*n* = 1171) [19]. However, subsequent studies also identified a female predominance in TGA patients [17,39]. A possible explanation could be the more frequent diagnosis of stroke mimics in women, despite similar symptoms as compared to men [40]. A recent review article also concluded that the age distribution of TGA cases appears to be similar to that of the general population [20]. To our knowledge, this study is the only one that performed an explicit comparison with public data from the registration register. It seems important to consider the age structure of the study population when comparing with the general population.

### 4.2. Cholesterol Levels

In previous studies, dyslipidemia was described either more frequently in stroke patients than TGA patients [18,41,42,43] or there were no differences between the groups [44,45,46,47,48]. In contrast, our study disclosed higher serum cholesterol levels on admission in TGA patients as compared to acute stroke patients [21]. Yet, ongoing lipid-lowering therapy, for example with statins, cannot be ruled out due to the retrospective study design. Overall, the data from the above-mentioned studies did not suggest an association between dyslipidemia and the occurrence of TGA [20]. As lipid-lowering therapeutics were not systematically available in our patients’ records, the increased cholesterol figures have to be discussed with caution. In addition, increased serum cholesterol levels (as compared to men) have already been described in postmenopausal women [49,50], with hormonal reasons being suggested for these sex differences. The assumption that the sex-specific differences shown are not specific to TGA is supported by the sex-specific difference shown in the control group of acute stroke patients.

### 4.3. C-Reactive Protein

C-reactive protein (CRP) [51] is a protein synthesised in the liver in response to inflammation. Measurement of high-sensitivity CRP (hsCRP) in serum/plasma has been proposed for assessment of cardiovascular disease risk profile in both men and women [52]. On average, the concentration of hsCRP in plasma is higher in women (by potentially up to 60%) compared to men with and without cardiovascular risk factors [51]. Corrected for age and cardiovascular risk factors, the different CRP figures in men and women of our study are attributable to the established sex differences, rather than to TGA occurrence. This is also supported by the trend of sex-specific differences shown in the control group of acute stroke patients.

### 4.4. DWI Lesions

A recent study reported higher numbers of TGA-associated DWI lesions in women compared to men [15], although the precise timing of MRI scanning in relation to TGA onset was not reported. In our study, no corresponding differences were encountered. The retrospective study design also restricted the precise definition in timing of TGA symptom onset and subsequent MR imaging. In the aforementioned study, the relevance of sex dependent differences in TGA numbers remained unclear and could not be explained on a pathophysiological basis.

### 4.5. Study Limitations

Unfortunately, a number of the reported TGA predictors, such as personal and family history of migraine, personal history of depression, positive family history of dementia, and a personal history of head injury, as well as several conditions under which TGA occurred (such as valsalva, medication, sexual intercourse) could not be investigated, due to the retrospective study design. In particular, these conditions were not systematically documented within patient records.

## 5. Conclusions

Our data enhance previous findings of sex-dependent differences in TGA patients by suggesting a relationship between precipitating events and increased blood pressure values upon admission in females. Furthermore, the association of augmented microangiopathy in female TGA patients with a higher frequency of TGA occurrence underlines the susceptibility to TGA episodes in the context of hypertensive episodes and their morphologic sequelae. The data may recommend that microangiopathic cerebral changes should be investigated, particularly in women with initial blood pressure emergency and occurrence of TGA. Furthermore, initial precipitating events should be more precisely ascertained to assess the possible association. The extent to which the short-term lowering of hypertensive blood pressure levels can reduce the risk of recurrent TGA is not yet known and could be subject to prospective studies.

## Figures and Tables

**Figure 1 jcm-11-05803-f001:**
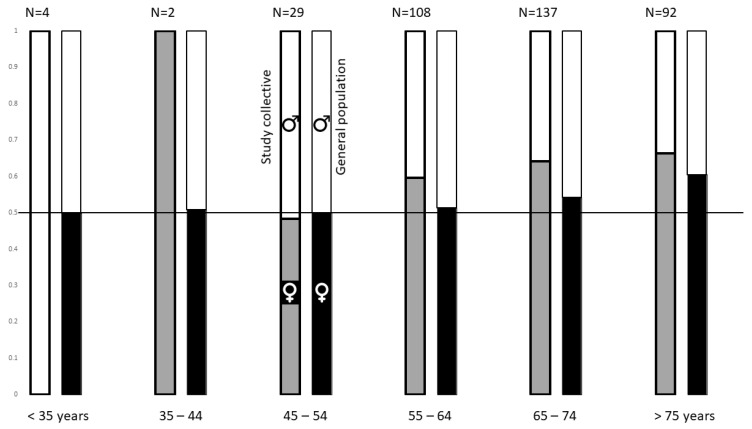
Grouped and stacked bar chart showing the relative sex distribution (y-axis) in different age groups (x-axis) in the study population (left column in each case) compared to the general population in the catchment area of the clinic according to public population data from 2019 (right column in each case) [25]. The left column in each case represents the relative distribution of women (grey, bottom) and men (white, top) among the study population in the indicated age group. The right column comparatively shows the relative frequency in the general population of women (black, bottom) and men (white, top). The horizontal line at height y = 0.5 indicates an equal sex distribution of 50% female. The absolute frequency of TGA patients in the corresponding age group is indicated above the study population bar (*n* = …). Caution is advised when reading the columns of the study population in the age group <35 years (*n* = 4, exclusively men) and the group 35–44 years (*n* = 2, exclusively women). Due to the small number of cases and exclusive inclusion of one group of sex, the column is only one colour in these cases.

**Figure 2 jcm-11-05803-f002:**
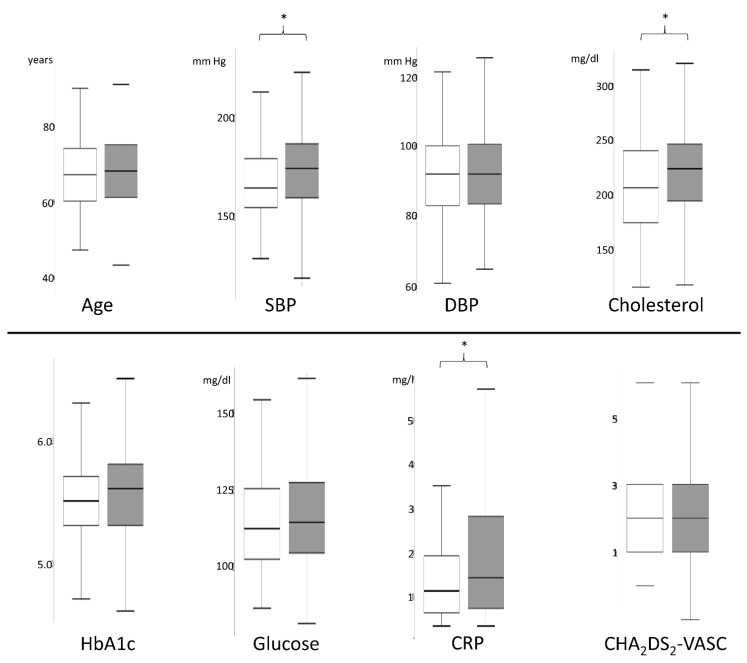
Comparison of the parameters of the vascular risk profile between female (grey) and male (white) TGA patients using boxplots showing higher systolic blood pressure, higher cholesterol levels, and higher C-reactive protein on admission in female TGA patients compared to male (* *p* < 0.05).

**Figure 3 jcm-11-05803-f003:**
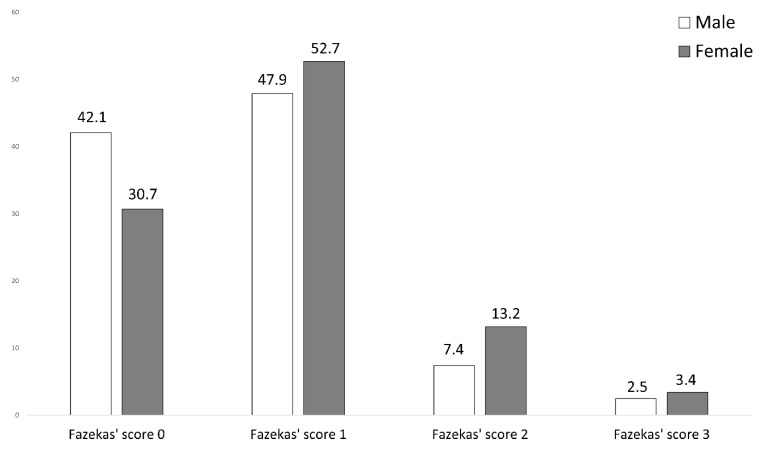
Graduation of cerebral microangiopathy using Fazekas’ score in female TGA patients (grey) compared to male TGA patients (white). Male TGA patients are more likely to have no microangiopathy, while microangiopathy is more common in female TGA patients. Y-axis: Distribution in percent. The numbers given denote the percentage of severity for the corresponding sex.

**Table 1 jcm-11-05803-t001:** Sex-specific comparisons of TGA patients (*n* = 372).

	Female TGA Patients (*n* = 230)	Male TGA Patients (*n* = 142)	Test Statistics
Age	68.1 ± 9.6	65.9 ± 11.4	U = 17,720.500, Z = 1.381, *p* = 0.167 ^c^
Hypertension	161/228 (70.6%)	101/140 (72.6%)	χ^2^ = 0.099, *p* = 0.753 ^a^
**Systolic blood pressure on admission**	**173.2 ± 23.4 mm Hg**	**165.8 ± 22.0 mm Hg**	**T = −2.725,** ***p* = 0.014 ^b^**
Diastolic blood pressure on admission	93.0 ± 13.0 mm Hg	92.3 ± 13.6 mm Hg	T = −0.407, *p* = 0.684 ^b^
Diabetes mellitus	12/228 (5.3%)	8/142 (5.6%)	χ^2^ = 0.024, *p* = 0.878 ^a^
Serum glucose level on admission	117.5 ± 19.2 mg/dL	116.3 ± 23.1 mg/dL	U = 14,335.500, Z = 1.257, *p* = 0.209 ^c^
HbA1c	5.6 ± 0.5 %	5.5 ± 0.7 %	U = 11,590.500, Z = 1.669, *p* = 0.095 ^c^
**Hypercholesterolemia** **(>200 mg/dL at admission)**	**140/202 (69.3%)**	**62/118 (52.2%)**	**χ^2^ = 8.994,** ***p* = 0.003 ^a^**
**Serum cholesterol level on admission**	**221.6 ± 40.7 mg/dL**	**207.6 ± 45.5 mg/dL**	**T = −2.800,** ***p* = 0.005 ^b^**
**CRP level on admission**	**2.8 ± 6.4 mg/L**	**1.7 ± 1.8 mg/L**	**U = 16,317.500,** **Z = 2.530,** ***p* = 0.011 ^c^**
LVEF < 50%	This comparison could not be computed, as only 3 TGA patients displayed LVEF < 50% (1 female, 2 male).		
Septal hypertrophy (male >10 mm, female > 9 mm)	66/96 (68.8%)	31/44 (70.5%)	χ^2^ = 0.041, *p* = 0.839 ^a^
Cerebral stenosis	17/210 (8.1%)	6/127 (4.7%)	χ^2^ = 1.414, *p* = 0.234 ^a^
Atrial fibrillation	21/228 (9.2%)	8/142 (5.6%)	χ^2^ = 1.550, *p* = 0.213 ^a^
CHA_2_DS_2_-VASc score (corrected for sex)	2.1 ± 1.5	2.1 ± 1.5	U = 16,022.500, Z = −0.312, *p* = 0.755 ^c^
Presence of DWI lesion	101/205 (49.3%)	61/122 (50.0%)	χ^2^ = 0.016, *p* = 0.898 ^a^
Unilateral vs. bilateral lesion in case of presence of DWI lesion	Bilateral 26/101 (25.7%)	Bilateral 13/61 (21.3%)	χ^2^ = 0.409, *p* = 0.523 ^a^
Antiplatelet therapy at discharge	129/229 (56.3%)	85/142 (59.9%)	χ^2^ = 0.447, *p* = 0.504 ^a^
OAC at discharge	19/229 (8.3%)	11/142 (7.7%)	χ^2^ = 0.036, *p* = 0.850 ^a^
Statin therapy at discharge	120/229 (52.4%)	77/142 (54.2%)	χ^2^ = 0.117, *p* = 0.732 ^a^
Antihypertensive drugs at discharge	159/229 (69.4%)	102/142 (71.8%)	χ^2^ = 0.242, *p* = 0.623 ^a^
Former stroke	28/229 (12.2%)	20/142 (14.1%)	χ^2^ = 0.268, *p* = 0.604 ^a^
Recurrence	19/230 (8.3%)	10/142 (7.0%)	χ^2^ = 0.181, *p* = 0.670 ^a^

^a^ Chi square, ^b^ parametric *t*-test, and ^c^ Mann–Whitney U-Test used as appropriate. Parameters highlighted in bold indicate significant differences between the sex-specific groups.

## Data Availability

The data that support the findings of this study are available from the corresponding author upon reasonable request. Statistical analysis of the data was carried out using Statistical Package for the Social Sciences (SPSS) version 25, 2018 (IBM, Armonk, NY, USA).

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
