# Peer review of "Transient Global Amnesia (TGA): Sex-Specific Differences in Blood Pressure and Cerebral Microangiopathy in Patients with TGA"

_jcm, 2022, doi:10.3390/jcm11195803_

Round 1

Reviewer 1 Report

In this study, the authors describe the sex differences in different clinical-radiological parameters among patients presenting with TGA. They show that female patients have higher systolic blood pressure, higher total cholesterol, higher CRP levels, and more pronounced cerebral microangiopathy. Overall, the manuscript is well-written and easy to follow. However, it is difficult to draw clinically meaningful conclusions. Given the relatively small sample size, the statistically significant differences may not have true significance in the real-world practice. For instance, there is a small, yet statistically significant difference in the systolic blood pressure (166 vs. 173). Additionally, without a comparison with an age-matched control group (without TGA), it is not possible to specifically attribute these differences to the acute presentation with TGA versus inherent sex differences alone. With that said, the conclusions of the study do not seem to be supported by the results. I have the follow suggestions for the authors:

1.       Consider a comparison with age-matched controls without TGA

2.       Consider performing a multivariable adjustment to eliminate the effect of potential confounders

3.       Most patients with TGA do not get evaluated with an echocardiogram, cerebral angiography, etc. It will be more important to incorporate the data on precipitating triggers and how they differed between males and females. 

Author Response

Thank you to this reviewer for the overall positive evaluation und the important suggestions. Please see here our response point by point:

  1. Indeed, we considered a control group for our analysis, but this is difficult to define. Due to the blood pressure abnormalities in TGA patients, healthy subjects may not be an appropriate control group. Since TGA appears clinically as stroke like symptom, we established acute stroke patients as a control group in a previously published study.

Significant different results of TGA patients (systolic blood pressure values, cholesterol levels, and CRP levels) were also analysed and compaired to 216 consecutive patients of our Stroke Unit with acute ischemic stroke or transient ischemic attack collected between November 24, 2018, and June 3, 2019 [22]. The inclusion criteria and demographic characteristics of the collective can be found in the cited publication and are not repeated here. The cited study showed after statistical correction for age that TGA patients had higher systolic blood pressure, higher cholesterol levels, and lower CRP levels than acute stroke patients. Supplementary analyses of these acute stroke patients were now carried out under the aspect of sex-specific differences of these parameters as a control condition of our significant parameters. The values for systolic blood pressure and cholesterol levels were not normally distributed, so the comparison was carried out using the Mann-Whitney U test.  The CRP values were normally distributed, so that the comparison was made using a parametric t-test. 

Analysis of acute stroke patients on admission with regard to sex-specific differences showed no differences of systolic blood pressure figures between the sexes (female stroke patients 162.6±24.5 mm Hg, male stroke patients 161.1±28.6 mm Hg, Mann-Whitney-U 4,565.000, Z=0.473, p=0.636). Serum cholesterol levels were higher in female stroke patients compared to male stroke patients (205.9±52.0 mg/dl versus 173.7±42.0 mg/dl; Mann-Whitney-U 7,162.000, Z=3.420, p=0.001). CRP levels showed a not significant trend to be higher in female stroke patients compared to male stroke patients (14.8±30.1 mg/l versus 8.2±19.1 mg/l; T=1.857, p=0.065).

In summary, while TGA patients presented sex-specific differences in systolic blood pressure values on admission, this was not the case in acute stroke patients. The sex-specific difference in cholesterol and CRP values in TGA patients can also be observed in acute stroke patients (at least for CRP values in the trend).

The distribution of the degree of cerebral microangiopathy was also analysed with regard to sex-specific differences in the cohort of acute stroke patients from our earlier published study [22]. Fazekas’ scoring excluded cerebral microangiopathy (Fazekas’ score 0) in 30.2% of female TGA patients (21.5% of male TGA patients), and disclosed microvascular findings as follows: mild microangiopathy (score 1) in 20.8% (35.4%), moderate microangiopathy (score 2) in 26.4% (20.0%), and severe microangiopathy (score 3) in 22.6% (23.1%). Using Mann-Whitney U test, the degree of microangiopathy (ranking from 0=none to 3=severe) was different between the sexes in acute stroke patients (U=1,691.500, z=0.173, p=0.862).

In summary, while TGA patients showed sex-specific differences in the distribution of the degree of cerebral microangiopathy, this was not the case in acute stroke patients.

  1. We considered performing multivariate analysis to adjust for potential confounders (i.e. age). This would have required parametric procedures such as multiple regression analysis or ANCOVA. However, these parametric procedures need a specific underlying data structure to produce statistically sound results, especially normally distributed and homoscedastic residuals. Both assumptions were severely violated in our data, indicated by significant Shapiro-Wilk tests and skewed q-q-plots.

While parametric procedures are usually considered robust against non-normal distribution in larger sample sizes (Norman, 2010), it is unclear what constitutes a “larger” sample size. Furthermore, the typical solution for this problem are outcome transformations that may cause biased estimates (Schmidt & Finan, 2018). Even if we ignored the normality assumption and considered all parametric procedures robust, we would still have to deal with heteroscedasticity. In his comparison of parametric MANOVA and non-parametric alternatives, Finch (2005) found that heterogeneity of (co-)variances still poses a serious threat with higher type I error rates and lower power than non-parametric alternatives.

We conclude with Osborne and Waters (2002) that there are “non-parametric statistical techniques available to researchers when the assumptions of a parametric statistical technique is not met. Although these often are somewhat lower in power than parametric techniques, they provide valuable alternatives”. In our study, this non-parametric alternative is the Mann-Whitney U test. The main disadvantage of this technique, multiple testing with alpha error accumulation, was dealt with by adjusting p-values with the Bonferroni method.

References:

Finch, H. (2005). Comparison of the performance of nonparametric and parametric MANOVA test statistics when assumptions are violated. Methodology, 1(1), 27-38.

Norman, G. (2010). Likert scales, levels of measurement and the “laws” of statistics. Advances in health sciences education, 15(5), 625-632.

Osborne, J. W., & Waters, E. (2002). Four assumptions of multiple regression that researchers should always test. Practical assessment, research, and evaluation, 8(1), 2.

Schmidt, A. F., & Finan, C. (2018). Linear regression and the normality assumption. Journal of clinical epidemiology, 98, 146-151.

  1. It is indeed relevant that precipitating factors seem to play a decisive role, whereby sex-specific differences of precipitating events have already been described in a recent study and this study has also been mentioned and discussed by us [Hoyer et al., citation 15].

Unfortunately, in our study a number of the reported TGA predictors, such as personal and family history of migraine, personal history of depression, positive family history of dementia, and a personal history of head injury, as well as several conditions under which TGA occurred (such as valsalva, medication, sexual intercourse) could not be investigated, due to the retrospective study design. In particular, these conditions were not systematically documented within patient records. A precise statement on sex-specific precipitating events is therefore unfortunately not possible with our data, which we have also mentioned in the study limitations.

Reviewer 2 Report

In this study, Rogalewski et al aim to analyze sex-specific differences among TGA patients in a large retrospective cohort from a tertiary referral center. There is no mention in their introduction/abstract of their recent study published in Frontiers, which aimed to determine predictors for TGA recurrence based on the recurrence rate in the same cohort including age and blood pressure. The current analysis does not appear to have taken the recurrence into account.

Although the aim of the paper may be relevant, there are several simplistic shortcomings in this study that make it difficult for it to support the authors' aim and not provide new findings as well.

The manuscript is difficult to read due to its poor writing style. Comments:

1) There is no mention of an aim in the abstract, which is poorly written. MRI is mentioned in the methods section but not in the results and conclusion section.

2) The study does not include a control group without TGA.

3) In their previous study, in Table 1 the authors compared TGA patients with and without subsequent recurrences (n = 340). However, I am uncertain whether the authors have considered the recurrence in their current study (Table 1).

4) The Discussion does not achieve its full potential. A Discussion should include an analysis of the results and clear, scientifically sound conclusions that are firmly supported by the data analysis. There should be a clear connection between the Discussion and the central hypothesis or question posed in the introduction. In other words, the Discussion should explain how the present evidence supports the main claims of the paper, or if it does not, why not. 

5) It is important to consider a multivariate analysis

6) The MRI results section is copied and pasted from a previous study.

7) The limitations of this study are the same as those of the previous study

Author Response

Thank you to this reviewer for the thorough review and the important comments. We respond here point by point and made the appropriate changes in the manuscript:

  1. As suggested, we have reorganized and rewritten the text of the abstract. The headings (background, methods, results, conclusion) were removed according to the requirements of the journal and the text was focused. Also, the aim was added as requested.

The results of the analysis with respect to TGA recurrence and the MRI findings are now clearly presented in the results section. Furthermore, the MRI findings are now also presented under conclusions.

  1. Indeed, we considered a control group for our analysis, but this is difficult to define. Due to the blood pressure abnormalities in TGA patients, healthy subjects may not be an appropriate control group. Since TGA appears clinically as stroke like symptom, we established acute stroke patients as a control group in a previously published study.

Significant different results of TGA patients (systolic blood pressure values, cholesterol levels, and CRP levels) were also analysed and compaired to 216 consecutive patients of our Stroke Unit with acute ischemic stroke or transient ischemic attack collected between November 24, 2018, and June 3, 2019 [22]. The inclusion criteria and demographic characteristics of the collective can be found in the cited publication and are not repeated here. The cited study showed after statistical correction for age that TGA patients had higher systolic blood pressure, higher cholesterol levels, and lower CRP levels than acute stroke patients. Supplementary analyses of these acute stroke patients were now carried out under the aspect of sex-specific differences of these parameters as a control condition of our significant parameters. The values for systolic blood pressure and cholesterol levels were not normally distributed, so the comparison was carried out using the Mann-Whitney U test.  The CRP values were normally distributed, so that the comparison was made using a parametric t-test. 

Analysis of acute stroke patients on admission with regard to sex-specific differences showed no differences of systolic blood pressure figures between the sexes (female stroke patients 162.6±24.5 mm Hg, male stroke patients 161.1±28.6 mm Hg, Mann-Whitney-U 4,565.000, Z=0.473, p=0.636). Serum cholesterol levels were higher in female stroke patients compared to male stroke patients (205.9±52.0 mg/dl versus 173.7±42.0 mg/dl; Mann-Whitney-U 7,162.000, Z=3.420, p=0.001). CRP levels showed a not significant trend to be higher in female stroke patients compared to male stroke patients (14.8±30.1 mg/l versus 8.2±19.1 mg/l; T=1.857, p=0.065).

In summary, while TGA patients presented sex-specific differences in systolic blood pressure values on admission, this was not the case in acute stroke patients. The sex-specific difference in cholesterol and CRP values in TGA patients can also be observed in acute stroke patients (at least for CRP values in the trend).

The distribution of the degree of cerebral microangiopathy was also analysed with regard to sex-specific differences in the cohort of acute stroke patients from our earlier published study [22]. Fazekas’ scoring excluded cerebral microangiopathy (Fazekas’ score 0) in 30.2% of female TGA patients (21.5% of male TGA patients), and disclosed microvascular findings as follows: mild microangiopathy (score 1) in 20.8% (35.4%), moderate microangiopathy (score 2) in 26.4% (20.0%), and severe microangiopathy (score 3) in 22.6% (23.1%). Using Mann-Whitney U test, the degree of microangiopathy (ranking from 0=none to 3=severe) was different between the sexes in acute stroke patients (U=1,691.500, z=0.173, p=0.862).

In summary, while TGA patients showed sex-specific differences in the distribution of the degree of cerebral microangiopathy, this was not the case in acute stroke patients.

  1. Indeed, we analysed sex differences in recurrent TGA in the present study. Unfortunately, Table 1 was presented on 2 pages and the information on recurrence was listed on the second page in the last row. Therefore, this reviewer may have missed this information.

In the results section (page 13, 1st paragraph), the analysis and the result of TGA recurrence is also presented. There was no difference between the sexes with regard to the recurrence rate. For better comprehensibility, we have listed the result from Table 1 in the text and referred to the table.

  1. According to the suggestions of this reviewer, we have completely re-organized, re-focused and re-written the discussion. Hereby, we believe, the reading improved, but even more important, the main findings and its discussion are in better balance and easier to understand.

  1. We considered performing multivariate analysis to adjust for potential confounders (i.e. age). This would have required parametric procedures such as multiple regression analysis or ANCOVA. However, these parametric procedures need a specific underlying data structure to produce statistically sound results, especially normally distributed and homoscedastic residuals. Both assumptions were severely violated in our data, indicated by significant Shapiro-Wilk tests and skewed q-q-plots.

While parametric procedures are usually considered robust against non-normal distribution in larger sample sizes (Norman, 2010), it is unclear what constitutes a “larger” sample size. Furthermore, the typical solution for this problem are outcome transformations that may cause biased estimates (Schmidt & Finan, 2018). Even if we ignored the normality assumption and considered all parametric procedures robust, we would still have to deal with heteroscedasticity. In his comparison of parametric MANOVA and non-parametric alternatives, Finch (2005) found that heterogeneity of (co-)variances still poses a serious threat with higher type I error rates and lower power than non-parametric alternatives.

We conclude with Osborne and Waters (2002) that there are “non-parametric statistical techniques available to researchers when the assumptions of a parametric statistical technique is not met. Although these often are somewhat lower in power than parametric techniques, they provide valuable alternatives”. In our study, this non-parametric alternative is the Mann-Whitney U test. The main disadvantage of this technique, multiple testing with alpha error accumulation, was dealt with by adjusting p-values with the Bonferroni method.

References:

Finch, H. (2005). Comparison of the performance of nonparametric and parametric MANOVA test statistics when assumptions are violated. Methodology, 1(1), 27-38.

Norman, G. (2010). Likert scales, levels of measurement and the “laws” of statistics. Advances in health sciences education, 15(5), 625-632.

Osborne, J. W., & Waters, E. (2002). Four assumptions of multiple regression that researchers should always test. Practical assessment, research, and evaluation, 8(1), 2.

Schmidt, A. F., & Finan, C. (2018). Linear regression and the normality assumption. Journal of clinical epidemiology, 98, 146-151.

  1. The presentation of the results of sex-specific differences in MRI imaging abnormalities (page 13 f.) has a paragraph on differences in the presence and laterality of hippocampal DWI lesions and a second paragraph on the extent of cerebral microangiopathy. Although the design of the reported results may be similar to previous studies, comparisons of these parameters have been made in previous studies between TGA patients and acute stroke patients, or between TGA patients with and without subsequent recurrence. In the current study, the same parameters (hippocampal DWI lesions, cerebral microangiopathy) were analysed in terms of sex differences and presented with the current results of this analysis. We have partly rewritten this paragraph and adapted it to this manuscript. Of course, there was no copy and paste of the previous results.

  1. It is correct that the current study was an extension of the cohort from previous studies [citations 6,22]. This is mentioned in the methods section in the "Patients" paragraph: "This investigation represents an extension of the cohort from previously reported studies [6,22]." Due to the expansion of the study population to include more patients in our clinic, but the same inclusion of patients, the limitations of the study are indeed the same.

In line with the reviewer's recommendation, we referred to our two previous studies at the end of the introduction and already pointed out there that the current study is an extension of the collective and an analysis with a different focus. By presenting the previous results, the differentiation between known and new results in our view is clearly clarified.
